# Predicting the Future Chinese Population using Shared Socioeconomic Pathways, the Sixth National Population Census, and a PDE Model

**Aijun Guo [1], Xiaojiang Ding [1], Fanglei Zhong [1,\*], Qingping Cheng [2,3] and Chunlin Huang [4,5]**

1   School of Economics, Lanzhou University, Lanzhou 730000, China
2   Northwest Institute of Eco-Environment and Resources, Chinese Academy of Sciences,
    Lanzhou 730000, China
3   University of Chinese Academy of Sciences, Beijing 100049, China
4   Heihe Remote Sensing Experimental Research Station, Northwest Institute of Eco-Environment and Resources,
    Chinese Academy of Sciences, Lanzhou 730000, China
5   Key Laboratory of Remote Sensing of Gansu Province, Northwest Institute of Eco-Environment and Resources,
    Chinese Academy of Sciences, Lanzhou 730000, China
*   Correspondence: zfl@lzu.edu.cn

**Abstract:** A precise multi-scenario prediction of future population, based on micro-scale census data and localized interpretation of global scenarios, is significant for understanding long-term demographic changes. However, the data used in previous research need to be further refined. Few studies have focused on predicting the sex ratio at birth, which is vitally important for estimating the future size and structure of the population. It is also important to interpret and set parameters for China's future population development in line with the framework for global shared socioeconomic pathways. This paper, therefore, used the structural population data for provinces, prefectures, and counties from the Sixth National Population Census of China. It comprehensively considered the impact of China's economic development level, specific population policies, and loss of an only child on key parameters, and localized the population change parameters for different scenarios. A population–development–environment model was used to explain the population change parameters. The population of 340 districts was refined, forecast, and aggregated to the national scale. The results show that the Chinese population is expected to first increase then decrease under the five paths from 2010 to 2050. The aging demographic structure is not reversed under any paths, and the increase or decrease in the urban and rural populations between adjacent node years is closely related to the fertility rate and urbanization speed. We suggest that measures should be taken to encourage childbearing, manage the aging population problem, and reduce the pressure on young and middle-aged people.

**Keywords:** population prediction; shared socioeconomic pathways; population–development–environment model; population structure; the Sixth Census; micro-scale census data; China

## 1. Introduction

Population figures are the foundation of human, social, and economic development. They are important in economic and social research disciplines, particularly for labor inputs and human capital investment [1] and are also the basis of other sustainable development evaluation disciplines, such as the study of natural resource consumption [2]. From a macro perspective, since China's reform and opening up in 1978, the country has made significant economic progress, the result of the demographic dividend brought about by the significant population base [3].

However, the large population, high fertility rate, and low mortality rate had a negative impact on the improvement of the living standards of Chinese people in the 1970s [4]. The government and scholars, therefore, agreed that population control was essential, and planned to achieve this through administrative, legal, and economic means. Since the 1980s, the government has successively introduced a policy encouraging one-child families, then a strict one-child policy and population control tightened [5]. The strict one-child policy led to a rapid decline in China's fertility rate. This fell from 5.7 in 1970 to 2.0 in 1990, below the normal replacement level of 2.1, and the total population was considered to have been controlled [6].

This had a wide range of economic and social effects. From a microeconomic point of view, only children had access to more resources, especially girls in urban areas, who had unprecedented power to resist gender stereotypes [7]. There were, however, also some unanticipated negative consequences [8]. Parents who had lost their only child suffered, and were often alienated from society [9–11]. From a macro perspective, the one-child policy failed to bring about normal population turnover. As a result, the previous significant demographic dividend will become a burden on the population over time. The accelerated aging of the population may expose China to the embarrassing situation of "being old before being rich" [12]. The two forces of discriminatory gender preference and the one-child policy, with access to gender-selective technology, disrupted the gender balance of the birth population. This will result in large numbers of single men, which poses hidden dangers for social security [13]. These circumstances are the background for the introduction of the "Comprehensive Two-Child" policy.

Figure 1 shows the total population and the number of births in China from 2000 to 2018. The total population of China has increased, but the number of births per year decreased from 2000 to 2004, and then showed a small overall increase. In 2015 to 2016, the number of births rose sharply before falling rapidly from 2016. Data from the National Bureau of Statistics of China in 2019 shows, that the total population of the mainland increased by 5.3 million in 2018 from the end of the previous year. The birth population was 15.23 million, but 2 million fewer than in 2017. This was the second year since the commencement of the "Comprehensive Two-Child" policy. The policy has helped to increase newborn numbers, which reached a peak of 17.86 million in 2016. Taking Shandong Province, which is generally considered to be "China's top province for children", as an example, the 2018 population data from cities show that the fertility rate has declined to varying degrees. The sluggish fertility rate and decrease in the number of births directly affect the population structure and other factors related to the healthy development of the Chinese population. They also have a major impact on economic, social, and environmental development. For example, they may affect the real estate market [14], promote capital deepening [15], and reduce the pressure on resources and environment [16,17]. It is important to understand the policy effects of different socioeconomic development policy preferences on the future population.

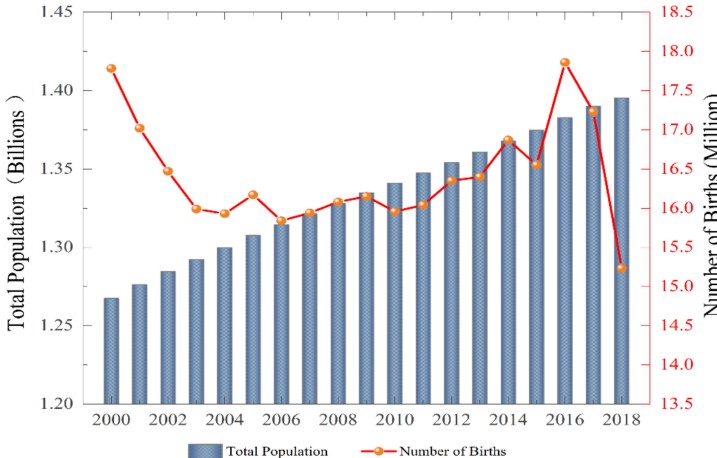

**Figure 1.** Total population and births in China from 2000 to 2018. (Data source: National Bureau of Statistics of China, http://www.stats.gov.cn/tjsj/.).

There are many studies on population prediction, using different methods. To predict the total population, Men et al. [18,19], Caimei et al. [20], Lai et al. [21], Folorunso et al. [22], Yan [23], Lee and Feng [24], and others have used grey prediction, neural network methods, and logistic and Malthus population models. These methods have provided valuable exploratory experience for population prediction by using a solid mathematical foundation, considering the general characteristics of population development or the inherent grey attributes of the population system. Other scholars have used methods that can predict the population structure, including the Leslie model [25] and the population–development–environment (PDE) model. The PDE model was developed by Lutz in 1994 [26]. He explained the philosophy behind the model in detail, arguing that any single dimension trend prediction that ignores other relevant factors must be divorced from reality. However, a model should be as concise and direct as possible to avoid 'black box' operations. In subsequent studies, Chinese scholars used the PDE model to predict the size and structure of China's population [27]. Some scholars have used this method to predict the population of particular areas [28] and specific age groups [29], which provides a reference for small-scale population predictions. In using these methods, scholars have also used many types of forecasting schemes, enriching the literature. However, these schemes were relatively simple, and they did not consider social, economic, and other factors.

The Intergovernmental Panel on Climate Change (IPCC) has developed an analytical framework of shared socioeconomic pathways (SSPs) to clarify population prediction in the context of climate change [30]. Elmar Kriegler and other scholars elaborated the concept of SSPs in detail, and included possible future developments and trends in key factors such as population, economies, social culture, conflict, technological progress, environment, and government management, making SSPs more useful for researchers [31]. O'Neill and other scholars provided detailed explanations of the key features and factors of SSPs, and developed comparative analysis on their basic and extended framework [32]. The framework evaluates the future social and environmental development paths qualitatively and quantitatively from the two dimensions of climate change mitigation and adaptation, and constructs five different development models [33]. These are SSP1 (sustainable development), SSP2 (moderate development), SSP3 (regional competition), SSP4 (unbalanced development), and SSP5 (fossil fuel-based development). The SSPs framework is widely used in many fields. The population predictions for the SSPs were initially carried out on a global scale [34] and for coastal areas [35]. Related studies about land use [36], population predictions [37], the magnitude of global warming [36] and the projection of global urban area growth [38] will assist in understanding the framework and application of SSPs. If SSPs are to be used in a small area, they must be scaled down [39,40]. Later, SSPs were examined by Chinese scholars and applied to China at the national and provincial levels [41,42], as well as to countries involved in the Belt and Road Initiative [43,44].

This article aimed to take into account as much as possible the heterogeneity of population structure between regions and the diversity of development paths, and clarify the future population development trends. It also aimed to provide input data to address questions about these demographic changes in China. We, therefore, attempted to improve the current population predictions in four ways. First, we comprehensively considered the impact of China's economic development level, its unique population policy, and other factors on the key parameters, such as the fertility rate. We localized the population change parameters for different economic and social development paths under the SSPs framework. This enabled us to link the population forecasts more closely with the social and economic development elements. Second, the basic data used in this paper were more elaborate than previous work, because we used structural data from the sixth national census for provinces, prefectures, and counties to improve the accuracy of the national population forecast. Third, we used a method of first dividing and then generalizing, using the characteristics of the PDE model, considering the population of the local area as a system, and then integrating it into the national picture to fully reflect the total population and structural characteristics of the population. Fourth, the accuracy of the model is quantified and compared with historical data, making the prediction more convincing.

## 2. Data and Methods

In China, the census from the National Bureau of Statistics provides the most detailed, systematic, and authoritative data on the population, including structural data by age and sex in small areas. The sixth census is the most recent data. Following the principles of systematization, integrity, effectiveness, scientific approach and operability, we used the data on provinces, prefectures, and counties from the sixth national census, including the number of men and women in 21 age groups, the total fertility rate of each province and autonomous region, and the average life expectancy of each province. The study covered 340 administrative units, including four municipalities and 336 prefecture-level and non-prefecture-level cities (collectively referred to as cities in this study), excluding the Hong Kong, Macao and Taiwan regions, and Chaohu, which merged into Hefei during the study.

We used the PDE model in this study. The accuracy and detail of any prediction depend on the model used. Some possible models were considered difficult to use for the purpose of this study. For example, those with "black box" characteristics, grey prediction or neural network methods make it hard to understand the prediction process. The ultimate saturation limit of the logistic model means that it cannot reflect changing trends in population development and the Malthus population model assumes a constant growth rate, which is obviously unreasonable in a long-term population forecast. The single-dimensional population projections made by these methods also do not reflect the structural changes within the population system, and particularly age structure, nor can they provide data about the future population, including the number of people by gender and age. The Leslie model defines the population fertility rate and mortality rate as constants, which fails to reflect the dynamic changes in these measures. We, therefore, used the PDE model as a way to overcome these shortcomings. The core idea of the PDE model is that the population should be seen as an ecosystem. The future population state depends on the quantity and structure of the existing population. The renewable capacity of the future population of a region is affected by the fertility rate, the mortality rate, and immigration and emigration of the population. The specific mathematical expression of the PDE model to calculate the population in a particular region is:

$$P_{(t+1,n+1)} = P_{(t,n)} \times (1 - D_{(t+1,n+1)}) + M_{(t+1,n+1)} \tag{1}$$

$$P_{(0,n+1)} = \sum_{t=15}^{49} [F_{(t,n+1)} \times FR_{(t,n+1)}] \times (1 - D_{(o,n+1)}) \tag{2}$$

$$F_{(0,n+1)} = P_{(0,n+1)} \times f\_r_{n+1} \tag{3}$$

$$P\_Z_{n+1} = \sum_{t=1}^{m} P_{(t,n)} + P_{(o,n+1)} \tag{4}$$

where t represents the age, n the year (n years), P the population, D the mortality rate, M the net migration of the population (net immigration is positive, and net emigration is negative), F the number of women, FR the fertility rate for specific age groups, f_r the proportion of women in the nascent population, P_Z the total population, and m the highest age of the population in the area.

## 3. The Combination of the PDE Model Parameters and SSPs Scenarios

### 3.1. Description of the SSPs Population Scenarios

The main parameters of the PDE model include the fertility rate, the mortality rate, and mobility rate. SSPs use the fertility rate, mortality rate, and education levels to describe the population. The two frameworks can, therefore, easily be connected to give a more complete and systematic future population development scenario. Table 1 summarizes the population characteristics in China under the SSPs, including fertility, mortality, life expectancy, and migration between urban and rural areas.

The relative scale of migration in and out of China is very small, and the population flow between different regions in China does not greatly affect the size or structure of the population as a whole. We, therefore, assumed that each prefecture was a closed region and that population migration was mainly the result of urbanization: That is, the transfer of the rural population to cities.

SSP1 describes an ideal state of sustainable development with low-carbon economic development, and normal population replacement. According to Maslow's hierarchy of needs, once their basic needs are met, people will find ways to meet the higher level needs, including for understanding, esthetic appreciation and purely spiritual needs [45]. As income increases, people will focus more on their quality of life, education, and health. As women's educational levels and economic independence increase, they are better able to balance family and work [17]. However, if a child dies, the one-child policy may affect people's sense of their place in the world [9]. Bereaved parents may cut themselves off from contact with friends and neighbors, leading to isolation [10,11]. Having two children may be a way to avoid this isolation in the event of a child's death [9]. Increased family income also makes people more inclined to have a second child where there is no mandatory population control [46,47]. Under SSP1, economic and social development is eco-friendly, and people generally have a high life expectancy. Overall, under this scenario, the population ecology (population age and gender structure) is good, population development involves a normal replacement level of fertility and a very low mortality rate and therefore a higher life expectancy. The compact and intensive urban form makes people's lives more convenient [48,49], and is also effective in improving the efficiency of energy and resource utilization. The urbanization rate is therefore faster.

SSP2 represents a medium-level development path. With the medium growth of income, people maintain a moderate level of health, education, and fertility, and the rural population gathers in cities at a medium speed.

SSP3 describes a fragmented social state in which economic growth is weak, various social contradictions are prominent, people are generally insecure and pessimistic about the future, population development is stagnant, and people are reluctant, do not want, or do not dare to give birth, so the fertility rate remains low [50]. Lack of income, anxiety, and nervousness lead to high mortality and reduced life expectancy. There are numerous social barriers, urban planning is ineffective, and people suffer from "urban diseases" [51]. The migration rate between urban and rural areas is slow.

SSP4 is a state of unbalanced development. China as a whole has a low fertility level, so it is assumed that cities will maintain a medium or low fertility level in the future, and the mortality rate will also be at a medium level. However, there is a slight difference in the migration of the urban and rural populations: High-income provinces have a moderate pace of urbanization [52], while people in low and middle-income provinces are rapidly moving from rural areas to cities to make a living.

Based on the development of fossil fuels, SSP5 has high carbon emissions and faces great challenges in mitigating climate change. In this situation, there is a focus on technological progress and investment in human capital. The development of the fossil fuel economy has provided a large amount of income. The fertility rate remains medium–high and the mortality rate is relatively low. The increase in wealth has made people more capable of moving from villages to towns, resulting in rapid urbanization.

**Table 1.** Population characteristics in China under the shared socioeconomic pathways (SSPs).

| Narratives | Fertility | Mortality | Life Expectancy | Migration Speed between Urban and Rural Areas |
|---|---|---|---|---|
| SSP1 | High | Low | High | Fast |
| SSP2 | Medium | Medium | Medium | Medium |
| SSP3 | Low | High | Low | Slow |
| SSP4 | Medium–low | Medium | Medium | High-income provinces, medium speed. Low-and middle-income provinces, fast |
| SSP5 | Medium–high | Low | High | Fast |

### 3.2. Calibration of Specific Population Parameters of SSPs

3.2.1. Total Fertility

The intermediate total fertility rate was mainly from the National Population Development Plan (2016–2030) issued by the State Council in 2016. This sets the national intermediate total fertility rate at 1.6 for 2015 and 1.8 for 2020, a level at which it then stabilizes. The national total fertility rates of low, medium-low, high, and medium-high levels were based on Chinese research [41,53], with the specific settings shown in Table 2. The total fertility rate of each prefecture was calculated based on the total fertility rate of the province in 2010 and the total national fertility rate. It was assumed that the fertility rate of women of all ages would remain at the 2010 level.

**Table 2.** Total fertility rates at different levels in China from 2015 to 2050.

| Total Fertility | 2015 | 2020 | 2025 | 2030 | 2035 | 2040 | 2045 | 2050 |
|---|---|---|---|---|---|---|---|---|
| Low | 1.52 | 1.632 | 1.632 | 1.54 | 1.54 | 1.54 | 1.54 | 1.54 |
| Medium-Low | 1.52 | 1.632 | 1.632 | 1.632 | 1.632 | 1.632 | 1.632 | 1.632 |
| Medium | 1.6 | 1.8 | 1.8 | 1.8 | 1.8 | 1.8 | 1.8 | 1.8 |
| Medium-High | 1.68 | 2.04 | 2.04 | 2.04 | 2.04 | 2.04 | 2.04 | 2.04 |
| High | 2.04 | 2.07 | 2.1 | 2.1 | 2.1 | 2.1 | 2.1 | 2.1 |

3.2.2. Mortality

For the national mortality rate in 2010, this study mainly used the "China Life Table for Life Experience of Life Insurance Industry (2010–2013)", issued by the China Insurance Regulatory Commission in 2016. At the provincial level, Wang and Ge's method was used [53]. This assumes that under medium conditions, the average life expectancy will increase by 0.15 years every year when the average life expectancy is between 70 and 75 years, by 0.1 years when the average life expectancy is 75–80 years, and by 0.06 years when the average life expectancy is over 80 years. We assumed that life expectancy at high (low) levels would be one year higher (lower) than medium levels every 10 years. The future life expectancy for each province was taken as the average of the life expectancy of the cities.

3.2.3. Migration Speed between Urban and Rural Areas

The migration of the urban and rural population was mainly based on predictions of China's urbanization level under the SSPs [42,54]. If the urban and rural population transfer speed was fast in the scenario, we set a fast urbanization speed (Table 1). To calculate the changes in urban and rural populations in a particular area and year, we used three steps. First, we considered the urban and rural areas of the region as a whole, and calculated its total population in a specific node year. Second, we used the total population and corresponding urbanization rate to calculate the corresponding urban and rural population sizes. Third, we used the size of the urban (rural) population in two adjacent node years to calculate the changes.

3.2.4. Sex Ratio of Births

According to Qian [55], Mu [13], and Chao et al. [56], when fetal sex selection technology is available, discriminatory gender preferences and the one-child policy disturb the gender balance of the birth population. This study, therefore, assumed that in future, the population policy will gradually be relaxed, that the public does not have strong gender preferences or undertake gender selection, and that the population will gradually return to the normal sex ratio. We, therefore, used a nonlinear fit based on the sex ratio of the newborn population from 2003 to 2017. When the sex ratio of the newborn population drops to the normal level of 102 boys for every 100 girls, we assumed that it

then remained unchanged. The specific formula is as follows, where y represents the sex ratio of the newborn population, x the year:

$$y = \begin{cases} \exp(14.96445 - 0.0050649x) & y > 102 \\ 102 & y \leq 102 \end{cases} \tag{5}$$

## 4. Accuracy of the Model Simulation

The accuracy of the population model simulation was measured by three indicators: the mean error (ME), the root mean squared error (RMSE), and the mean absolute relative error (MARE). These indicators are calculated as:

$$\mathrm{ME} = \frac{1}{n}\sum_{i=1}^{n} (m_i - \hat{m}_i) \tag{6}$$

$$\mathrm{RMSE} = \sqrt{\frac{1}{n}\sum_{i=1}^{n} (m_i - \hat{m}_i)^2} \tag{7}$$

$$\mathrm{MARE} = \frac{1}{n}\sum_{i=1}^{n} (|m_i - \hat{m}_i|/m_i) \tag{8}$$

where $m_i$ represents the i-th actual value, $\hat{m}_i$ the i-th predicted value, and n the number of data.

Considering the availability and comparability of population data, we used the total populations of the provinces in 2015 published by the National Bureau of Statistics as the comparison object for the predicted population. Based on the detailed population data in the sixth census, this study compared the total population of each province in 2015 under SSP2 with the data published by the National Bureau of Statistics. Figure 2 shows the specific comparison. After the calculation, the ME was 3.4, the RMSE 149.8, and the MARE 0.0289, which shows an ideal simulation accuracy.

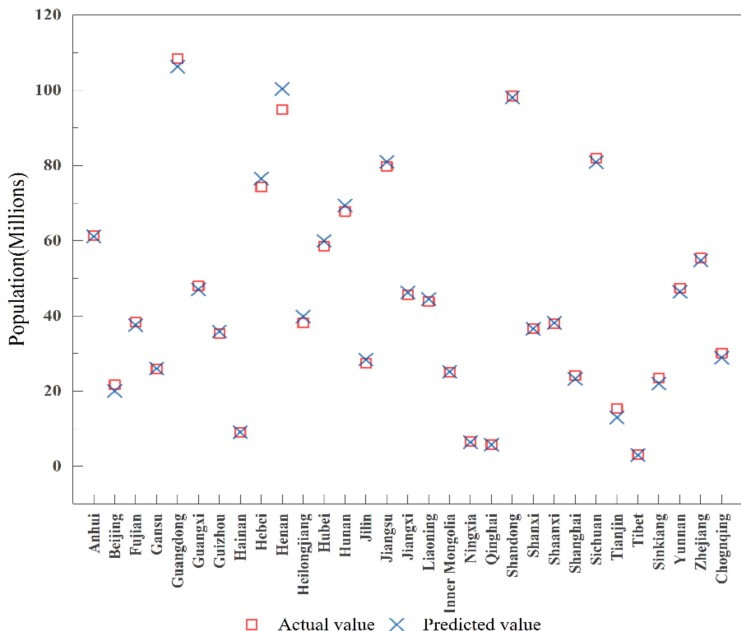

**Figure 2.** Comparison of predicted and actual values of the total population of the provinces.

## 5. Results

### 5.1. Total Population

The total population of China under the SSPs is shown in Figure 3. There are two commonalities in the development of China's population under the five paths. First, the population under the SSPs will initially show an increase, followed by a decrease. Second, the turning point of the total population occurs around 2035. The difference in China's total population under the different paths is mainly reflected in the peak population and the extent of change. Under SSP2, the population will grow from 1.33 billion in 2010 to 1.458 billion in 2030. It will reach a peak of about 1.46 billion in 2035 and then fall to 1.38 billion in 2050. The World Population Prospects in 2017 suggested that it would reach 1.36 billion by 2050 under the medium-variant projection [57]. The World Population Prospects in 2019 suggested that China's population is expected to rise to 1.46 billion in 2030, then fall to 1.40 billion in 2050 [58]. In the National Population Development Plan (2016–2030) [59], the population of China is expected to reach 1.45 billion in 2030. China's population development under our intermediate forecast is, therefore, comparable to the forecasts made by other institutions. Of the five paths, the population is expected to be highest under SSP1, reaching a peak of about 1.5 billion people in 2035, falling slightly to 1.46 billion by 2050, but still a large population base. The populations under SSP3, SSP4, and SSP5 will reach 1.436 billion, 1.441 billion, and 1.49 billion, in 2035 and decline after that. By 2050, the populations under SSP3 and SSP4 will be relatively close, and both below 1.36 billion.

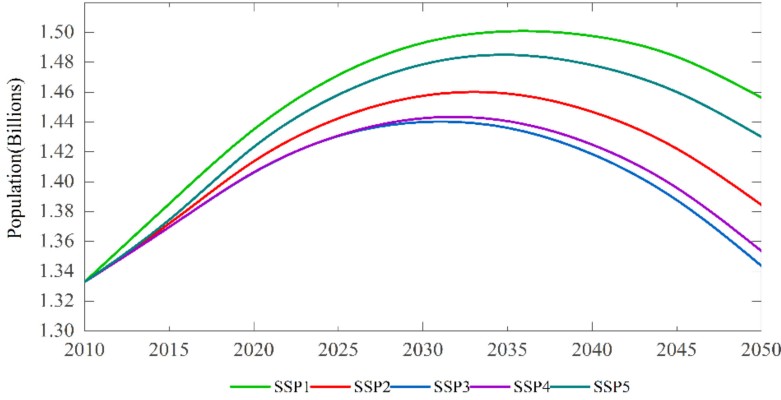

**Figure 3.** Predicted total population of China under SSPs from 2015 to 2050.

### 5.2. Age Structure

Figure 4 shows the age structure of China's population, with a "spindle-shaped" population pyramid under all five paths by 2035. The bottom and top of the towers will be pointed, and the majority of the population will be between 25 and 84 years old, with the largest number in the group aged 45–49 years. This is older than the current age structure. The tower bottoms under each path are similar. However, the population structure under the SSPs will shrink in 2050, with the population over 60 years old accounting for more than half of the total population. Comparing the pathways, the bottom of the population pyramid will be narrower under both SSP3 and SSP4, and China will face the dilemma of having few or no children. Under SSP1 and SSP5, the distribution of the young population at all ages will be relatively uniform, and the population ecology will be rebuilt. The population under SSP2 will be in the transition period of population development, and the total population aged 0–14 years will decrease slightly from the oldest to the youngest.

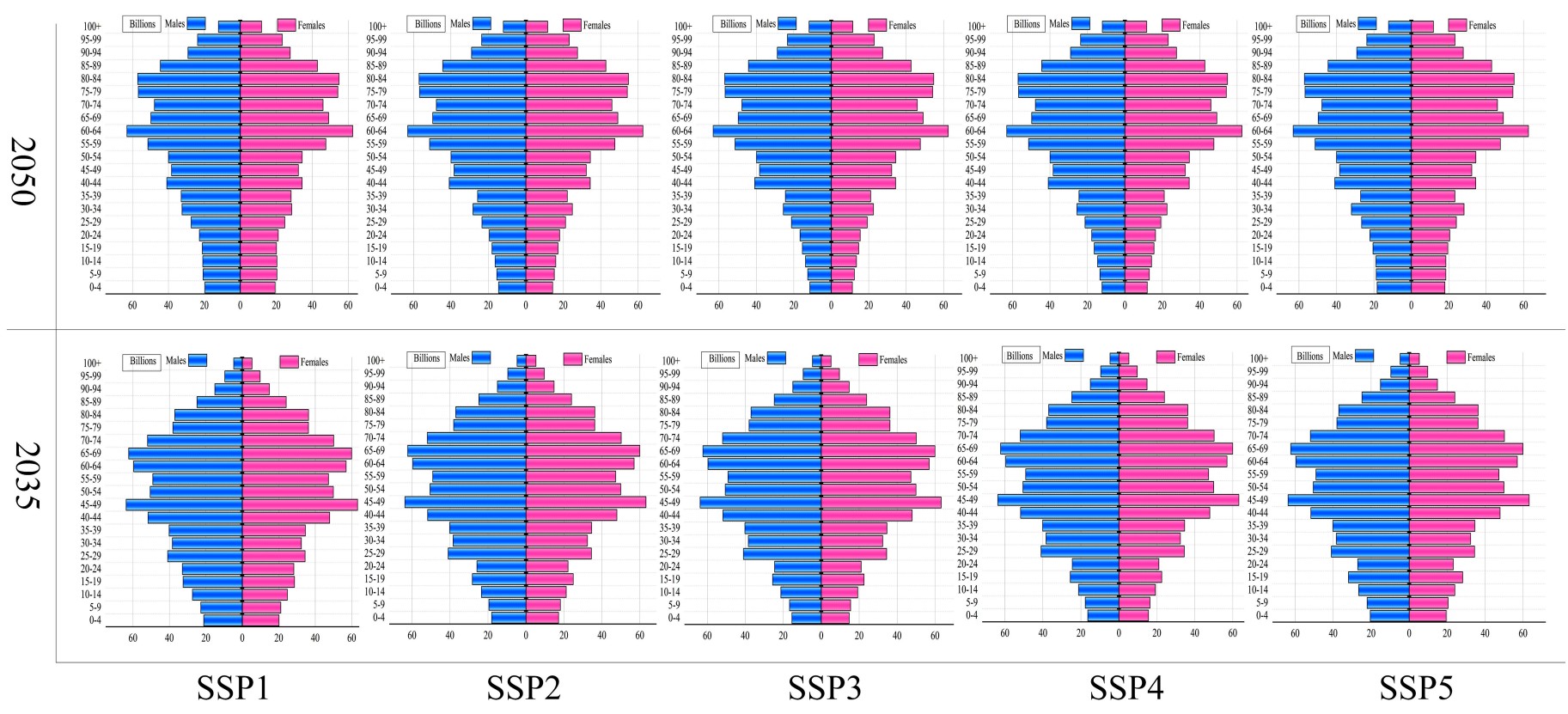

**Figure 4.** Chinese population pyramid under the SSPs in 2035 and 2050.

The population was further broken down into five age groups: 0–14, 15–59, 60–69, 70–79, and over 80 years old. Those over 60 years old were further divided into three groups, younger (60–69 years), moderate (70–79 years), and older (80 years and over) [60]. The population development of each age group from 2010 to 2050 is shown in Figure 5.

In all five scenarios, the population aged 15–59 will shrink dramatically, from 70% in 2010 to less than 40% in 2050. From 2010 to 2050, the precise number of people aged 15–59 varies by SSP scenario. SSP1 shows the smallest decrease, of about 355 million people, which is the smallest decrease of the five scenarios. Under SSP3, the population of this group will experience the largest reduction, of about 418 million people. Under SSP2, SSP4, and SSP5, there will be decreases of about 396 million, 415 million and 371 million, respectively.

The numbers aged over 60 will gradually increase under all five paths, mainly among the younger group (60–69 years) and the over 80s. In 2010, the proportions of the younger, moderate, and older groups in the total population were 7.49%, 4.26%, and 1.58%. By 2050, there will be smaller differences between these proportions under all five scenarios, with the middle group accounting for about 14% of the total population, and the younger and older groups accounting for about 15% and 23%. Figure 5 also shows that the proportion of older people will exceed half of the total population by 2050.

The number of people aged 0–14 will decline in the future. In 2010, 16.6% of the population was aged 0–14 years, but by 2050, this will be less than 10% under all of the SSP paths. Under the moderate SSP2 scenario, about 6.7% of the population will be aged 0–14 years in 2050. Even if the population reaches normal replacement levels in the future, only about 8% will fall into this age group in 2050. Under the relatively pessimistic SSP3, the proportion will only be about 5%. Under the two extreme scenarios of SSP1 and SSP3, population aging is irreversible.

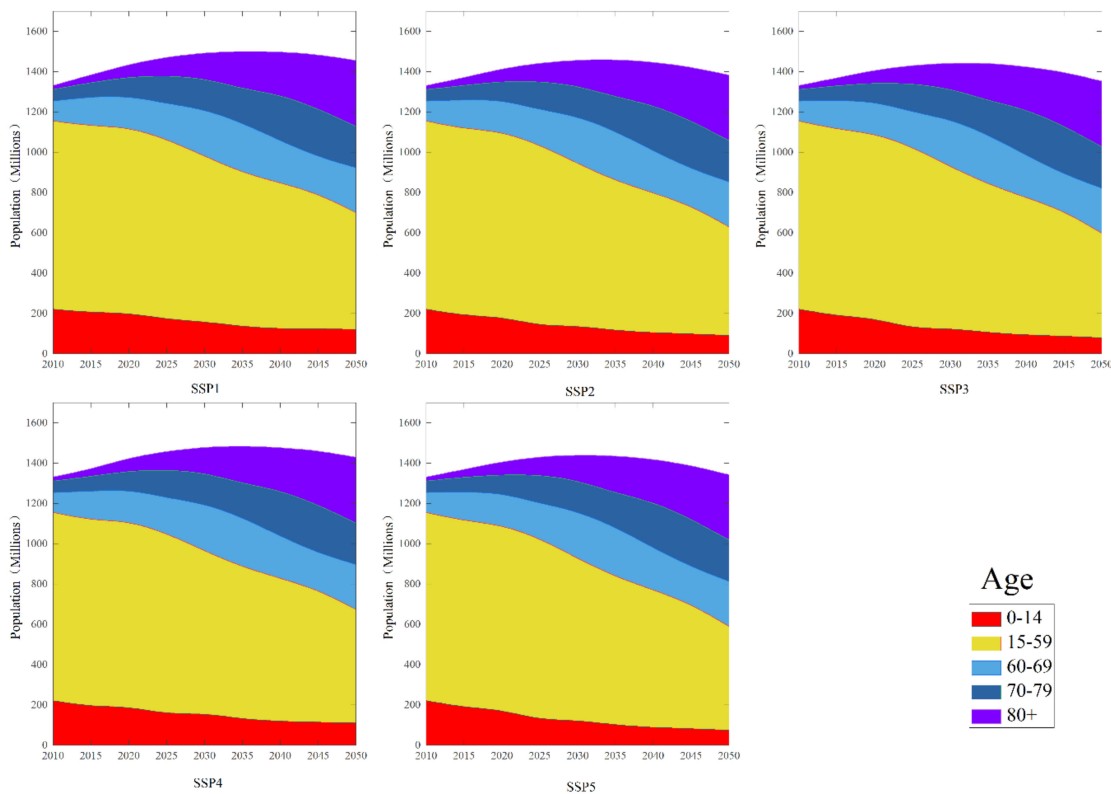

**Figure 5.** Population of China for 2010–2050 by age groups under the SSPs.

### 5.3. Dependency Ratio

The dependency ratio measures, to some extent, the influence of the age structure on the economy and society. The indicator can be subdivided into child dependency and old-age dependency. Generally,

the population aged 0–14 is referred to as the population of children, the 15–64 age group is the working population, and the population aged 65 and above is older people. The formulas for calculating dependency ratios are:

$$r_{0-14} = \frac{pop_{0-14}}{pop_{15-64}} \tag{9}$$

$$r_{65+} = \frac{pop_{65+}}{pop_{15-64}} \tag{10}$$

where r represents the dependency ratio and pop the population.

Figure 6a shows the child and old-age dependency ratio in the Chinese population under the SSPs for 2010–2050. Under each path, the child dependency ratio will continue to decrease from the 2010 rate of 0.22 until around 2040. It will rise slightly after that, but remain less than 0.2 until 2050. In stark contrast, the old-age dependency ratio will continue to rise from 0.12 in 2010 to a high of around 0.9 in 2050. Under SSP1, there is a relatively high child support ratio, and the old-age dependency ratio is the lowest of the five scenarios. The child dependency ratio is at its lowest, and the old-age dependency ratio at its highest under SSP3. The scenario under SSP4 is similar, except that the fertility rate is higher, reducing the aging of the population to some extent. The fertility rate must rise to a certain level to have any effect on the old-age dependency ratio. It is not accurate to measure the social burden by directly comparing the populations of different age groups if the different ages groups have different consumption abilities. Following Chen et al. [61], this paper, therefore, made a weighted comparison between children and older people to obtain a weighted dependency ratio. The formulas for calculating these dependency ratios are:

$$r\_w_{0-14} = \frac{pop_{0-4} \times 0.25 + pop_{5-14} \times 0.45}{pop_{15-64}} \tag{11}$$

$$r\_w_{65+} = \frac{pop_{65+} \times 0.7}{pop_{15-64}} \tag{12}$$

where r_w represents the weighted dependency ratio and pop the population.

The results calculated for the weighted dependency ratio are shown in Figure 6b. The child and old-age dependency ratios are similar under all five paths. Overall, the child dependency ratio, which was 0.085 in 2010, will start to decline, reaching a level of around 0.05 in 2050. The old-age dependency ratio will rise from 0.08 in 2010 and reach a level just below 0.7 in 2050. Taking into account the consumption power of the different age groups, the social dependency ratio will be reduced, particularly the burden associated with supporting older people.

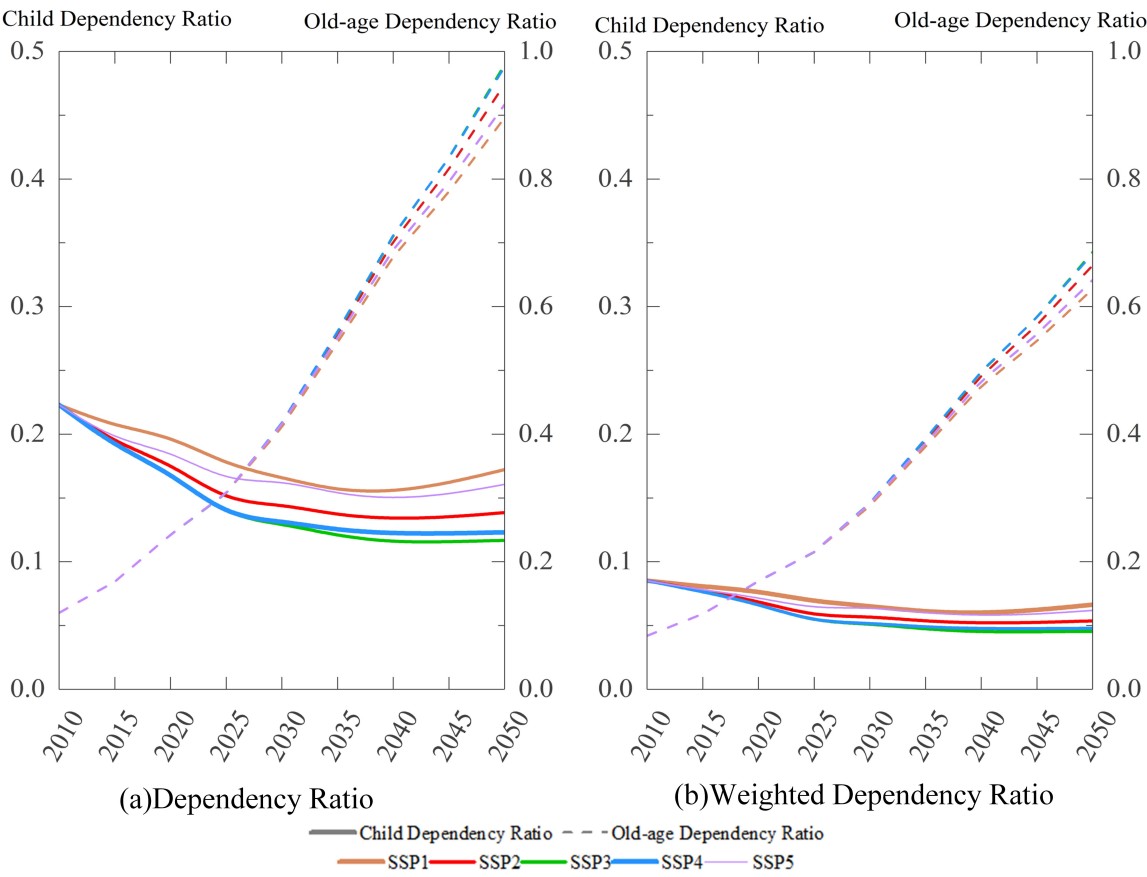

**Figure 6.** Dependency ratios for China for 2010–2050 under the SSPs.

*5.4. Urban and Rural Populations*

The increase or decrease in the urban and rural populations between adjacent node years in China from 2015 to 2050 is shown in Figure 7. On the whole, the population in cities is expected to increase, but the rate of increase will slow each year. The rural population will fall every year. The increase (decrease) in the populations of cities (villages) varies between scenarios. Under SSP1, the fertility rate is high, the new population is large, and the urbanization speed is relatively fast. The population shift from rural to urban areas is largest under this scenario, with the increase ranging from 155 million in 2015 to 9 million in 2045. SSP5 will be similar. Under SSP4, the shift from rural to urban areas will be slightly lower. Under SSP3, the population's fertility rate is low, the rate of urbanization is slow, and the number of farmers accepted by towns, and population leaving rural areas, are at their lowest. However, by 2045, the shrinkage of the urban population will exceed the transfer of the rural population to towns under SSP3. Both urban and rural areas face declining populations under this scenario. As a medium-development scenario, SSP2 has the second lowest population of the five paths. By 2050, the increase in the urban population under all the SSPs will be close to negative, and the rural population will decrease by 19 million to 32 million than in 2045.

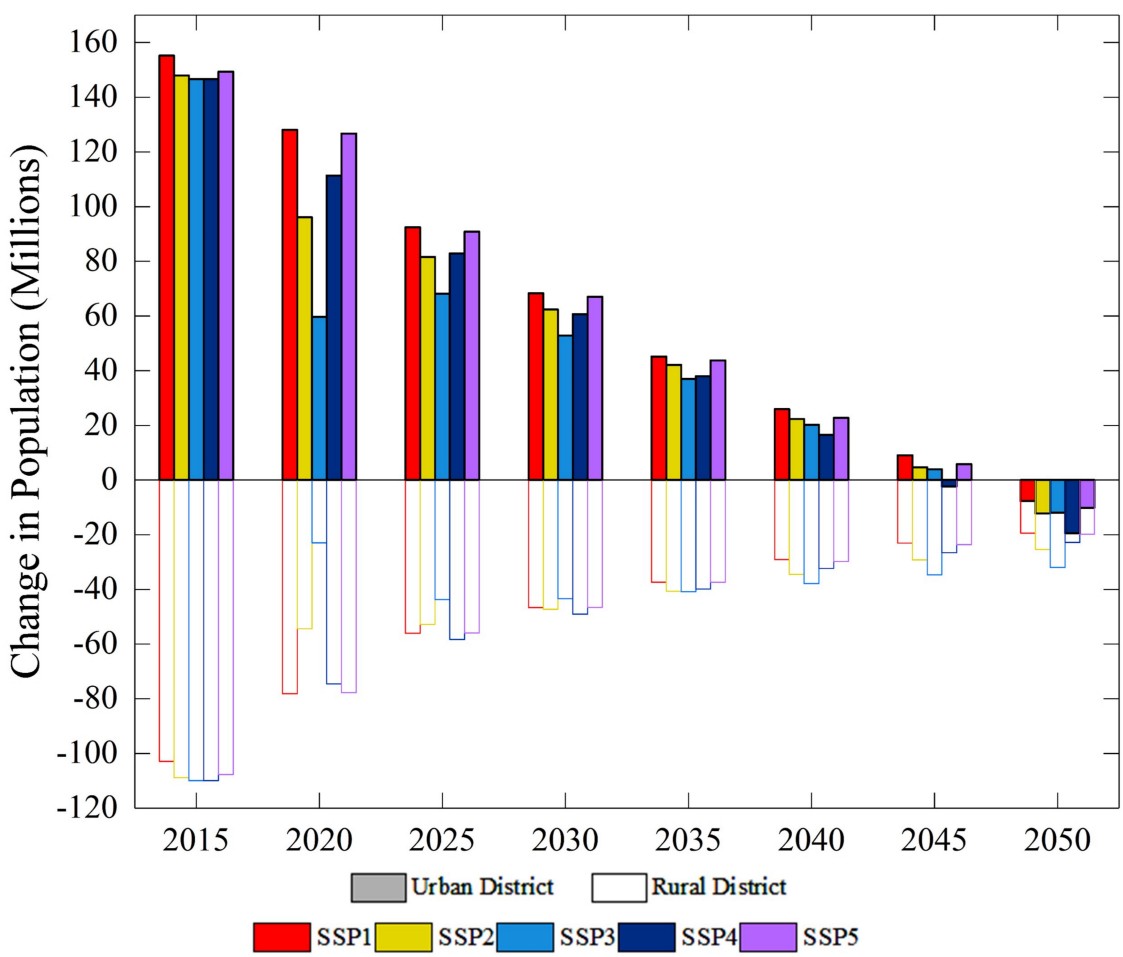

**Figure 7.** Increase or decrease in the urban and rural populations in China from 2015 to 2050 under the SSP scenarios.

## 6. Conclusions and Policy Recommendations

### 6.1. Conclusions

This study used structural demographic data for cities from the sixth census, a PDE model and the SSP development scenarios to aggregate the population of 340 regions in China into a national-scale population. A number of conclusions can be drawn.

1.  China's population is expected to reach a peak in 2035 and will continue to decline until 2050. By 2050, the population will be largest under the SSP1 scenario, at 1.46 billion people. Under SSP2, with moderate development, the total population will be 1.38 billion. Under SSP3 and SSP4 the population will be close to 1.35 billion. Under SSP5, the population will be higher than under SSP2 but lower than under SSP1.

2.  Under the five paths, China's population age structure in 2050 will be "spindle-shaped", showing an aging population. Even under the optimistic SSP1, with a higher newborn population, this aging will remain difficult to reverse. Under other scenarios, the pressure will be greater and society will face the situation of fewer or even no children.

3.  The population aging will not be reversed in the future. Under all five scenarios, the population aged 15–59 will shrink significantly, from 70% in 2010 to less than 40% in 2050. The decrease in population is smallest under SSP1 and largest under SSP3. The proportion of older people will gradually increase under all five paths, particularly the groups aged 60–69 years and over

80 years. The number of people aged 0–14 will also decrease in the future, falling below 10% by 2050 under every scenario.

4.  Increasing the fertility rate could reduce population aging to a certain extent, provided it exceeds a threshold, or the effect on the old-age dependency ratio will be very small. Under each path, the child dependency ratio will drop from 0.22 in 2010 to less than 0.2 in 2050. The old-age dependency ratio will rise from 0.12 in 2010 to a high of about 0.9 in 2050. The relationship between the child-rearing ratio and the old-age dependency ratio in each path involves a trade-off. When the spending power of the different age groups is taken into account, the social dependency ratio is greatly reduced. In particular, the burden of supporting older people will be reduced.

5.  The increase or decrease in the urban and rural populations is closely related to the fertility rate and urbanization speed. With the exception of SSP3, the urban population in all scenarios will increase, although the rate of increase will slow, and the rural population will continue to decline every year.

## 6.2. Policy Recommendations

The policy recommendations merging from our analysis focus on the need to encourage childbearing and to liberalize the restrictions on population control. Faced with a widespread situation of people "not wanting to have children", "not being able to have children" and "not being able to raise children", the government should do everything possible to increase sources of income and reduce expenditure for families. It should encourage childbearing by rewarding those who have children, and by implementing comprehensive, family-friendly social policies on maternity leave, good prenatal and postnatal care, education, and medical care, to create an environment that encourages and supports parents across society. This will involve respecting individual wishes to give birth, and removing the restrictions on childbearing to give families the right to choose how many children to have.

The Chinese government should improve the old-age support system and actively respond to the aging crisis [62]. First, policies should enable the increasingly large group of older people to live healthy lives, by improving the provision of medical services for this group and vigorously promoting knowledge of old-age care. Second, to ensure the quality of life and make older people happy, it would be helpful to organize literary and art activities or create platforms to encourage and develop older people's interests and hobbies. Finally, we should tap into the dividends of the aging population, and encourage younger people to value the concept of "doing something for older people", and older people to create value, and generally make the lives of older people more enjoyable.

We should actively address the shortage of a young, strong labor force and ease the pressure on the young and middle-aged to support older people. Considering the size of China's population of older people in the future, it may be helpful to encourage delayed retirement, while respecting individual wishes [63], to take advantage of the greater experience of older people and also relieve some of the economic pressure on young people. It is also necessary to improve the skills of workers and to develop intelligent machines to improve working efficiency and ensure the normal operation of the economy.

## 7. Discussion on Uncertainty

There are inevitably some uncertainties about the population forecasts in this paper. First, the data in the sixth population census are still flawed, and contain some omissions and registration errors. However, there is no authoritative correction for these data at present. Second, it remains an open question whether the fertility rate in each region will reflect the fertility rate in the whole country in 2010. Third, the mortality rate among older people will be affected by education levels, living habits, and medical standards, but it is not clear whether the mortality rate will be affected by the size of the population. Finally, the urbanization of the population will be affected by new policy factors, including

the construction of the Belt and Road Initiative, the coordinated development of regional economies, and the implementation of the rural revitalization strategy.

**Author Contributions:** Conceptualization, F.Z.; methodology, X.D.; resources, C.H.; writing—original draft preparation, X.D.; writing—review and editing, F.Z. and Q.C.; visualization, X.D. and Q.C.; supervision, A.G. and F.Z.; project administration, F.Z.; funding acquisition, F.Z.

**Funding:** This research was funded by the Strategic Priority Research Program of the Chinese Academy of Sciences (Grant No. XDA19040500), National Natural Science Foundation of China (Grant No. 41571516), and the Fundamental Research Funds for the Central Universities (Grant No. 2019jbkyjb013).

**Acknowledgments:** We greatly appreciate the positive and constructive comments from the anonymous reviewers and the editors.

**Conflicts of Interest:** The authors declare no conflict of interest.

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
