# Peer review of "Predicting the Future Chinese Population using Shared Socioeconomic Pathways, the Sixth National Population Census, and a PDE Model"

_sustainability, doi:10.3390/su11133686_

Round 1
Reviewer 1 Report
The article has a potential but some elements need improvements:
· There is a need to add some information where authors start to analyze the figures and tables
· For some readers can be not clear what IPCC means.
· Please try to consider if the presentation of the limitation of the applied model in the introduction is proper idea? I suggest the chapter 2 or after results.
· Objective of the article should be formulated clearly.
· There is to less information about one child policy and two child policy. The first had impact on the population changes in the past, and the second one would have in the future. There any assumption related to that in the model? There is also any information that Census data can include some errors for example children born but not registered – the author should add information that those data cannot be very precious – Maybe there are any research about the scale of unregistered population in China?
· The assumption about fertility – the author should explain why these assumption based on old data were used (from 2010 when one child policy was obligated)
· The subchapter 3.2.3 is too short – when subchapter is distinguished there is believe that some information would be presented – so I suggest to present data
· In the subchapter 3.2.4, formula 5 there is no explanation what e means
· In the subchapter 5.3 there in information “Figure 5(a) shows” it is not clear whether it is firth graph on figure 5 or there is mistake and should be 6(a)?
Reviewer 2 Report
The paper considers the impact of China’s economic development level, specific population policies and the pain of losing the only child on key parameters and localizes the population change parameters for different economic and social development SSP paths.
In the study a population/development/environment (PDE) model is used to explain the population change parameters.
In my opinion, the paper is interesting and is also easy to read.
However, it must be improved with reference to three aspects:
1. why do the authors adopt the illustrated methodology? There are also other approaches, with respect to which nothing is said and no reciprocal advantages or disadvantages are highlighted;
2. the bibliography is modest and must be integrated;
3. the issue examined in the article has multiple effects, of a social, cultural, environmental and economic nature. I suggest mentioning this aspect. In this regard, I suggest reading and considering for references:
Bencardino M., Nesticò A., Demographic Changes and Real Estate Values. A Quantitative Model for Analyzing the Urban-Rural Linkages. Sustainability 2017, Vol. 9, Issue 4, 536, doi: 10.3390/su9040536. MDPI AG, Basel, Switzerland.
Round 2
Reviewer 2 Report
The authors have improved the article, making the necessary additions